

# Leaf defenses of subtropical deciduous and evergreen trees to varying intensities of herbivory

Xiaoyu Liu[1,2], Carri J. LeRoy[3], Guobing Wang[4], Yuan Guo[1,2], Shuwang Song[1,2], Zhipei Wang[1,2], Jingfang Wu[1,2], Fenggang Luan[2], Qingni Song[1,2], Xiong Fang[5], Qingpei Yang[1,2], Dongmei Huang[6] and Jun Liu[1,2]

[1] Jiangxi Province Key Laboratory for Bamboo Germplasm Resources and Utilization, Jiangxi Agricultural University, Nanchang, China
[2] Jiangxi Agricultural University, Nanchang, China
[3] Evergreen State College, Olympia, WA, USA
[4] Department of Scientific Research, Administration of Jiangxi Guanshan National Nature Reserve, Yichun, China
[5] College of Land Resources and Environment, Jiangxi Agricultural University, Nanchang, China
[6] School of Humanities and Public Administration, Jiangxi Agricultural University, Nanchang, China

Corresponding author
Jun Liu, liujun@jxau.edu.cn

## ABSTRACT

Generally, deciduous and evergreen trees coexist in subtropical forests, and both types of leaves are attacked by numerous insect herbivores. However, trees respond and defend themselves from herbivores in different ways, and these responses may vary between evergreen and deciduous species. We examined both the percentage of leaf area removed by herbivores as well as the percentage of leaves attacked by herbivores to evaluate leaf herbivore damage across 14 subtropical deciduous and evergreen tree species, and quantified plant defenses to varying intensities of herbivory. We found that there was no significant difference in mean percentage of leaf area removed between deciduous and evergreen species, yet a higher mean percentage of deciduous leaves were damaged compared to evergreen leaves (73.7% *versus* 60.2%). Although percent leaf area removed was mainly influenced by hemicellulose concentrations, there was some evidence that the ratio of non-structural carbohydrates:lignin and the concentration of tannins contribute to herbivory. We also highlight that leaf defenses to varying intensities of herbivory varied greatly among subtropical plant species and there was a stronger response for deciduous trees to leaf herbivore (*e.g.,* increased nitrogen or lignin) attack than that of evergreen trees. This work elucidates how leaves respond to varying intensities of herbivory, and explores some of the underlying relationships between leaf traits and herbivore attack in subtropical forests.

## INTRODUCTION

Insect herbivory on trees is a common phenomenon in forests, by both specialist and generalist herbivores (*Price, 1991*; *Barone, 1998*). Leaves are tissues with high nutrient status, are easily consumed by insects, and provide a rich resource for many consumers (*Schowalter, Hargrove & Crossley, 1986*). Insect herbivores cause leaf area loss and tissue

damage, which influence plant photosynthesis and growth (*Zhang & Turner, 2008*; *Kerchec et al., 2012*; *Visakorpi et al., 2020*), and can affect community biodiversity (*Huntly, 1991*; *Jefferies, Klein & Shaver, 1994*). In tropical forests, it has been shown that leaf herbivores influence interspecific competition and vegetation community structure (*Hairston & Hairston, 1993*; *Coley & Barone, 1996*; *Kurokawa & Nakashizuka, 2008*). Leaf herbivory levels in subtropical forests may be different than in tropical forests, due to variation in both plant community structure and insect herbivores, but there are few studies of subtropical forests for comparison.

Generally, leaf traits determine the preference and choice of leaf herbivores, including nutrient status, chemical defense substances and other physical properties (*Gómez et al., 2008*; *Loranger et al., 2012*). Some studies indicate that insects prefer feeding on palatable leaves with higher concentrations of nutrients, a pattern which is outlined in the plant vigor hypothesis (*Price, 1991*; *Cornelissen, Fernandes & Vasconcellos-Neto, 2008*; *Che-Castaldo et al., 2019*). Indeed, secondary metabolites (*e.g.*, tannins) in leaves could decrease leaf palatability and increase defensive capabilities (*Harborne, 1991*; *Bennett & Wallsgrove, 1994*). The role of secondary metabolites in defense may involve herbivore deterrence, anti-consumer activity, and tissue toxicity, which can all deter further herbivore damage (*Bennett & Wallsgrove, 1994*; *Kazakou et al., 2019*). Therefore, we suspected that insect herbivores would consume more palatable leaves with higher concentrations of nutrients and non-structural carbohydrates as well as lower concentrations of defensive compounds (*e.g.*, tannins and lignin) (Hypothesis 1).

Evergreen and deciduous species vary greatly in terms of leaf longevity and phenology, which influence concentrations of structural carbohydrates (*e.g.*, lignin, cellulose) and non-structural carbohydrates (NSC; *e.g.*, starch and sugars) (*Palacio et al., 2007*; *Michelot et al., 2012*). Indeed, leaf longevity correlates with some key traits associated with growth and resource-uptake rates (*Wright et al., 2004*). For example, compared to thinner deciduous leaves, evergreen species tend to have thicker leaves with a longer leaf-life span (*Withington et al., 2006*) and lower nutrient requirements due to their lower annual leaf construction costs (*Ye, Kitayama & Onoda, 2022*). Therefore, we hypothesized that deciduous tree leaves would suffer worse leaf herbivore attack with both a higher percentage of leaf area damaged and a higher percentage of leaves damaged due to their suite of leaf quality traits (Hypothesis 2). Here, we examined both the percentage of leaf area removed by herbivores as well as the percentage of leaves attacked by herbivores to evaluate leaf herbivore damage across 14 subtropical deciduous and evergreen trees. Using these two separate measures of leaf herbivory has been suggested as an approach to better demonstrate variable plant responses to herbivory (*Beck & Labandeira, 1998*; *Maldonado-López et al., 2019*).

Herbivores can reduce plant performance and fitness. In turn, plants use tolerance and resistance strategies to prevent or mitigate herbivory impact and maximize fitness (*Schuman & Baldwin, 2016*; *Matthias, 2018*; *Salgado-Luarte et al., 2023*). Under herbivore attack, plants mount a defense response characterized by the accumulation of secondary metabolites and inhibitory proteins (*Kerchec et al., 2012*). Empirical evidence has demonstrated that there can be a trade-off between defense and growth and that investment in defense may result in reduced loss to herbivores, but also reduced resources for growth

or reproduction (*Moore et al., 2003*; *Fine et al., 2006*). Plants have evolved a wide diversity of antiherbivore defenses to variable herbivore pressures due to differential costs and investments, such as constitutive and induced defenses. Such induced responses are likely a cost-saving strategy compared to constitutive defense (*Karban, 2020*), and some scholors believe that specialist herbivores may cause distinct induction of plant defenses compared with those induced by generalists (*Poelman et al., 2008*; *Zhang & Turner, 2008*; *Ali & Agrawal, 2012*).

Condensed tannins are very important phenols and chemical defense substances in plants, and can delay, reduce or deter the amount of herbivore damage (*Agrawal, 2011*; *Karban, 2011*). However, such chemical resistance is assumed to be relatively costly (*Stamp, 2003*; *Zhang et al., 2018*; *Cipollini & Heil, 2010*), especially the production of high concentrations of tannins. Previous studies have shown different responses of nutrients and chemical defense to varying intensities of leaf herbivory because of their various leaf traits, such as leaf thickness and toughness (*Hanley et al., 2007*). For example, there can be decreased nutrients and NSC in attacked plant tissues, because they are diverted away from the site of damage and into storage tissues (*Newingham, Callaway & BassiriRad, 2007*; *Gómez et al., 2010*; *Quijano-Medina et al., 2019*). Additionally, plants can produce more secondary metabolites to provide effective anti-herbivore resistance (*Karban, 2020*). Therefore, we hypothesized that leaf herbivore attack would decrease the concentrations of nutrients and NSC and produce more tannins for defense, particularly under heavy herbivory (Hypothesis 3).

To test the above three hypotheses, we analyzed herbivore damage on 14 tree species including both deciduous and evergreen trees in secondary subtropical forests. We quantified plant defense outcomes to varying intensities of herbivory, and analyzed the relative contributions of plant nutrient status (*e.g.*, concentration of nutrients and nonstructural carbon) as well as defensive traits (*e.g.*, concentration of tannins, lignin and cellulose) to leaf area removed. Overall, the present work builds towards linking damage to putative responses through ambient (natural) levels of herbivory in subtropical forests.

## MATERIALS AND METHODS

### Study site

Portions of this text were previously published as part of a preprint (https://www.researchsquare.com/article/rs-1949227/v1). We conducted this study in a secondary forest at Guanshan National Nature Reserve in South China (28°30′∼28°40′N; 114°29′∼114°45′E). The region has a subtropical warm and humid climate, and the mean annual air temperature is 16.2 °C. The mean annual precipitation is 1,950∼2,100 mm, which mainly occurs from March to August (*Liu & Zhang, 2017*). In 2014, we established a 12 ha biodiversity monitoring plot, completed a full plant catalogue, and determined the location of all woody plants with diameter at breast height (DBH) >1 cm. All field monitoring observations for this study were conducted in this plot, and were permitted by the local government of the aforementioned sites. Field experiments were approved by the local government of Jiangxi Guanshan National Nature Reserve (project number: The National Natural Science

Foundation of China, 32060319). In this area, there were two vertical strata: the first is composed of deciduous trees of rapid growth and some older evergreen species that reach a maximum height of 15–20 m and form an upper canopy. The second stratum is composed of a dense understory with lianas, adult trees of shade-tolerant species and juvenile trees.

### Leaf herbivory

We selected fourteen common woody plant species (including seven deciduous and seven evergreens) according to species and community composition in this forest and six individuals with equal DBH and height from each species ($n = 6$) (Table S1), and measured their percentage of insect herbivory in August 2019. The sampling was conducted in the second half of the growth season, when the majority of insect herbivores had completed their feeding but well before the start of leaf fall, which is common in leaf damage studies (*Thakur et al., 2021*; *Kozlov, Zverev & Zvereva, 2022*). We sampled trees from across the 12 ha plot according to their mean DBH and canopy height of each species in this area (Table S1). All selected trees were sampled from similar terrain and altitude to ensure similar habitats and herbivory environments. We collected four branches (the diameter and and length were almost 1 cm and 80 cm, respectively) in the middle canopy from the four cardinal directions of each individual, then picked up and mixed leaves as one independent sample, excluding some leaves that were entirely consumed or prematurely abscised due to damage. We scanned all leaves using a plant image analyzer (Hangzhou WSeen Detection Technology Co., Ltd, Hangzhou, China) and measured total leaf area as well as the area of leaf damaged by insect herbivores. In total, almost 9,000 leaves from 84 individuals of the 14 study species were scanned and analyzed.

For each plant, percentage of leaf area removed by herbivores was calculated as: ((removed area/total area) ×100%) and the percentage of leaves attacked was calculated as: ((the number of insect attacked leaves/total number of leaves) ×100%). For leaf area removed, we divided all sampled leaves into four categories: unattacked (0% leaf area removed), mild herbivory (<10% leaf area removed), moderate herbivory (10–50%) and heavy herbivory (>50%) (*Thakur et al., 2021*). Such categorical assessment of leaf damage is common in leaf damage studies and provides an overview of the severity of insect herbivory on plants (*e.g.*, *Schuldt et al., 2010*; *Zverev, Zvereva & Kozlov, 2017*). To avoid contamination of plant samples because of microbes on the leaf surfaces and prevent the loss of some active ingredients due to enzymatic reactions before performing chemical analyses, all samples were cleaned with ultra pure water and heated to 105 °C for 30 min to kill all micro-organisms, then oven-dried at 65 °C to constant weight, and ground into a fine powder using a ball mill (JX-2010, Shanghai, China) for chemical analysis. There were a total of 336 plant samples (14 species × 4 leaf herbivory levels × 6 replicates) in this experiment.

### Chemical analyses

The concentrations of total C and N were measured on 2 mg subsamples of leaf powder using an elemental analyzer (IR-MS; Thermo Fisher Scientific, Waltham, MA, USA) with acetanilide as external standard. All C and N measurements were run in duplicate and the

average deviations of replicate analyses from the means were 1.1% for N and 0.2% for C concentrations. To extract P in samples, 150 mg of dried plant powder from each sample was digested using $H_2SO_4$ and $H_2O_2$ at 350 °C with a microwave digestion system for 30 min, and then washed with deionized water and diluted up to 100 mL (final volume). Prior to analysis, all digested samples were then analyzed following the molybdenum antimony method and a molybdate–ascorbic acid procedure using ultraviolet visible (UV-Vis) spectrophotometry (880 nm, UV-5100), and calibrated using a standard solution (*Lu, 2000*). NSCs were extracted using an enzymatic digest and UV spectrophotometry methods, and analyzed as described in *Wong (1990)* and *Hoch (2007)*. The standard anthrone colorimetric method was employed to measure the contents of plant NSCs (*Li et al., 2016*; *DuBois et al., 1956*). Briefly, 100 mg of ground leaf sample was placed in a 50-mL centrifuge tube and mixed with 10 mL 80% (v/v) alcohol, extracted in 90 °C in hot water for 10 min three times. All the extracted solution was transferred into a 50 mL flask and we adjusted the final volume to 50 mL for the measurement of soluble sugar *via* the anthrone colorimetric method. The residue after three extractions in the centrifuge tube was dissolved using 30% (V/V) $HClO_4$ for 12 h, then extracted in 80 °C hot water for 10 min. After that, the extracted residue was cooled down, filtered, and diluted to 50 mL in a flask for the determination of starch. Soluble sugars and starch contents were calculated as a fraction of the dry matter of the leaf (mg $g^{-1}$) respectively (*Li et al., 2016*; *Xie, Yu & Cheng, 2018*).

To extract lignin, 200 mg of dried plant powder from each sample was digested with ferrous ammonium sulfate hexahydrate ($H_8FeN_2O_8S_2 \cdot 6H_2O$) and CuO in a high-pressure reactor (LS-150LD, China). The reactor was heated to 140° C, and the mixture was stirred for 3 h, the 20 µL 5.49 mM ethyl vanillin was added and the sample was rinsed with ultrapure water. The flow rate was set to 0.6 mL $min^{-1}$ and undiluted samples of 10 µL were injected and analyzed on a HPLC (SCIEX, Shanghai, China) (*Sluiter et al., 2008*). To extract cellulose, 200 mg of dried sample was digested with a mixture of acetic acid and nitric acid, and was titrated with ammonium ferrous sulfate solution. To extract hemicellulose, 200 mg of dried sample was digested with hydrochloric acid and determined in reduction of copper and iodine (*Lu, 2000*). The concentrations of condensed tannins were determined using the Folin Ciocalteu assay. To extract tannins, 200 mg of dried plant powder from each sample was digested in a water bath at 80 °C, then a 100 µL aliquot of extract was added to 750 µL of distilled water, 500 µL Folin-Ciocalteu reagent and 1,000 µL of 35% sodium carbonate ($Na_2CO_3$). The mixture was shaken vigorously after diluting to 10 mL with distilled water. The mixture was incubated for 30 min at room temperature and read at 725 nm using a UV-Vis spectrophotometer. A gallic acid (GA) standard curve was prepared for 0–200 mg/L and tannin content and was expressed in mg gallic acid equivalent/g dry matter. The total tannin content was expressed as gallic acid equivalents (GAE)/g dry matter, as calculated from the prepared standard curve with 0–100 mg/ GA (*Tamilselvi et al, 2012*). All chemical analyses were run in duplicate. Additionally, we calculated the ratios of C:N, NSC:lignin, and NSC:cellulose as indicators of leaf quality.

### Leaf defense to herbivory

We calculated leaf defense to herbivory (LDH) at the level of individuals by comparing the concentrations of nutrients and defense compounds in different levels of damaged and undamaged (or rarely damaged) leaves. Here, varying LDH could indicate different sensitivity or defense of leaves to insect herbivory among species (*Thakur et al., 2021*). We measured defense outcomes to determine if a tree species increased (positive values) or decreased (negative values) production of a compound under varying levels of herbivory. We calculated the LDH for each compound, N, P, NSC, tannins, lignin, and cellulose, using the equation below:

LDH=1−(content in herbivore-attacked leaves/content in undamaged or rarely damaged leaves)×100%

where content = concentrations of N, P, NSC, tannins, lignin, or cellulose, and herbivore-attacked leaves including mild, moderate, heavy herbivory. We calculated leaf defense to herbivory (LDH) by comparing the concentrations of nutrients and defense compounds in different levels of damage to undamaged leaves at the level of individuals.

### Data analysis

We determined the concentrations of nutrients (N, P and NSC) and defensive compounds (tannins, lignin and cellulose) in all samples (including damaged (mild, moderate, heavy herbivory) and undamaged leaves), then compared these traits between deciduous and evergreen trees, among tree species, and leaf defense across varying levels of herbivory compared to undamaged or rarely damaged leaves. Differences in leaf traits between deciduous and evergreen species were detected using Student's $t$-tests (two-tailed test). We analyzed the effect of leaf traits on the percentage of leaf area removed using generalized linear mixed models (GLMMs). The percentage of leaf area removed was set as a response variable and the concentration of tannin, lignin, cellulose, NSC and the ratio of NSC: Lignin, NSC: Cellulose in un-attacked leaves were included as explanatory variables, and the effect of the species as included as a random effect. We measured the relative contribution of variables to percentage of leaf area removed from multiple regression models using LMG analysis by the R-package relaimpo (2022). Alpha = 0.05 was the criterion for significant differences for all tests. Statistical analyses were performed using SPSS 24.0 (IBM, SPSS 24.0., Chicago, USA) for Windows, and graphs were created using Origin (OriginLab, Northhampton, MA, USA).

## RESULTS

### Leaf herbivory on evergreen and deciduous trees

Overall, there were significant differences in leaf herbivory between deciduous and evergreen subtropical tree species as a whole, and among species within each plant type. Mean percentage of leaf area removed by herbivores ranged from 4% to 19% for deciduous trees, and from 2% to 12% for evergreen trees (Fig. 1A). There was no significant difference in mean percentage of leaf area removed between deciduous ($n = 7$, 10.0%) and evergreen species ($n = 7$, 8.2%) ($t = -1.754$, $P_{2-\text{tailed}} = 0.084$) (Fig. 1A'). However, the percentage of leaves attacked by herbivores ranged from 47% to >90% for deciduous species and from

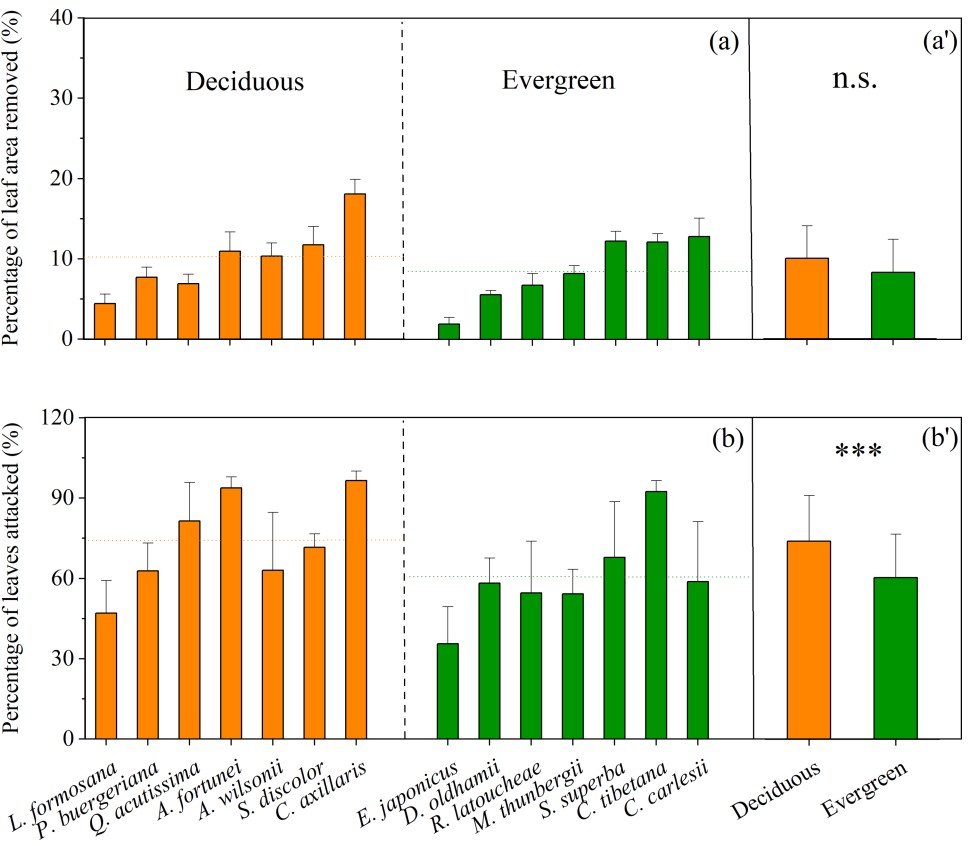

**Figure 1** **Mean (±1 SE) percentage of leaf area removed by herbivores (A, A') and the percentage of leaves attacked by herbivores (B, B') in the Guanshan National Nature Reserve for leaves of 14 study species in 2020.** Both deciduous (orange, $n = 7$) and evergreen (green, $n = 7$) species were included: *Liquidambar formosana, Padus buergeriana, Quercus acutissima, Alniphyllum fortunei, Acer wilsonii, Sapium discolor, Choerospondias axillaris*, and *Elaeocarpus japonicus, Daphniphyllum oldhamii, Rhododendron latoucheae, Machilus thunbergii, Schima superba, Castanopsis tibetana, Castanopsis carlesii.* Dotted lines indicate the mean percentage of leaf area removed (A') and the mean percentage of leaves attacked (B') for both deciduous and evergreen species. Significant differences between deciduous and evergreen species are denoted with ***$P < 0.001$, n.s $P > 0.05$.

35% to 90% for evergreen species (Fig. 1B), with a higher mean percentage of deciduous leaves damaged (73.7%) compared to evergreen leaves (60.2%, Fig. 1B', $t = -3.413$, $P_{2-tailed} = 0.001$).

## Correlation between leaf herbivory and leaf traits

In our study, we found that the leaf concentrations of nutrients in deciduous leaves were higher than those of evergreen species, including N, P, NSC, and the ratio of N:Lignin, NSC:Lignin, and NSC:Cellulose. In contrast, we found higher concentrations of hemicellulose and C:N in deciduous leaves (Table S2), which indicated that deciduous leaves were more palatable than evergreen leaves to some extent. Moreover, we found that the percentage of leaf area removed was negatively correlated with NSC:lignin and hemicellulose, and positively correlated with the concentration of tannins, N, and cellulose

**Table 1** The relative contributions (% of the variation explained) of leaf traits (N, cellulose, C, hemicellulose, tannin, NSC:lignin) in terms of explaining the percentage of leaf area removed by herbivores.

| Multiple regression model | | Variables | Estimate | Std. error | P | Relative contribution (%) |
|---|---|---|---|---|---|---|
| | | N | 0.0271 | 0.006 | <0.0001*** | 6.0 |
| *P* value | <0.0001 | Cellulose | 0.0036 | 0.001 | <0.0001*** | 6.9 |
| | | C | −0.0014 | 0.001 | 0.068 | 8.1 |
| | | Hemicellulose | −0.0092 | 0.002 | <0.0001*** | 13.7 |
| Adj R$^2$ | 0.2939 | Tannin | 0.0166 | 0.005 | <0.0001*** | 28.7 |
| | | NSC:Lignin | −0.3292 | 0.050 | <0.0001*** | 36.7 |

**Notes.**
In this multiple regression model, the *P* value and Adj R$^2$ indicate overall model statistics. Estimates >0 indicate an expected positive relationship between percentage of leaf area removed and leaf traits. Estimates <0 indicate an expected negative relationship between percentage of leaf area removed and leaf traits.
*** Significance level: P<0.001.

(Table 1). We found that there were no significant effects of leaf traits on the percentage of leaf area removed, including the concentrations of tannin, lignin, cellulose, NSC and the ratio of NSC: Lignin, NSC: Cellulose (Table 2).

## Leaf defenses to varying intensities of herbivory

When comparing nutrient concentrations in attacked *vs.* unattacked or rarely damaged leaves from the same species using the leaf defense to herbivores (LDH) calculation, we found some interesting variation both among species, and between deciduous and evergreen species. The difference between the N content in herbivore-attacked leaves compared to undamaged leaves tended to be higher (more negative) as herbivore intensity increased, but stronger for most deciduous species, particularly in medium and high intensity herbivory (Fig. 2A, $t = 9.673$, $P_{2-tailed} = 0.045$; $t = −2.744$, $P_{2-tailed} = 0.024$). These results suggest that there is a stronger response of N to insect herbivores in deciduous species. Similarly, for three of seven deciduous species, significantly lower concentrations of phosphorus were found for herbivore-attacked leaves (Fig. 2B). Patterns are more mixed when examining differences in NSC across intensities of herbivory. In this case, low or medium levels of herbivory resulted in significantly lower concentrations of NSC for two of seven deciduous species (*P. buergeriana* and *A. fortunei*, Fig. 2C). For evergreen species, low levels of herbivory were related to significantly lower concentrations of NSC for *D. oldhami*, *S. superba*, *C. tibetana*, and *C. carlesii* (Fig. 2C).

For defensive traits of tree leaves (*i.e.,* the concentrations of tannins, lignins and cellulose), we found that medium or high intensity herbivory (>10%) was associated with higher concentration of foliar tannins for three of seven decidous species and four of eight evergreen species (Fig. 3A). For all other species, there were indications that insect herbivory did not significantly stimulate the production of foliar tannins (Fig. 3A). The concentration of lignin was stimulated at high levels of herbivory for almost all deciduous and evergreen species (Fig. 3B). Interestingly, the patterns for cellulose are opposite, with most species showing lower concentrations of cellulose, especially under low or medium levels of herbivory intensity (Fig. 3C).

**Table 2 The generalized linear mixed models between percentage of leaf area removed and leaf traits.**

| Generalized linear mixed model | | | | Variables | $F$ | $P$ |
|---|---|---|---|---|---|---|
| df1 | df2 | $F$ | $P$ | | | |
| | | | | Tannin | 0.419 | 0.520 |
| | | | | Lignin | 3.178 | 0.080 |
| 6 | 53 | 3.244 | 0.009 | Cellulose | 2.974 | 0.090 |
| | | | | NSC | 3.876 | 0.054 |
| | | | | NSC: Lignin | 1.624 | 0.208 |
| | | | | NSC: Cellulose | 1.985 | 0.165 |

**Notes.**

In this generalized linear mixed model, the df1, df2, $F$ and $P$ indicate overall model statistics, including degrees of freedom (df), $F$ and significance ($P$).

Significance level: $^*P < 0.05$.

$^{**}P < 0.01$.

$^{***}P < 0.001$.

$^{n.s}P > 0.05$.

# DISCUSSION

## Leaf - herbivory performance and preference in subtropical forests

In our study, the mean overall percentage of leaf area removed was only 8.2%–10% in these subtropical tree species, which is lower than that seen in other forests. It has been reported that the mean overall leaf damage at the individual level across all forests is 14% (*Martini & Goodale, 2020*). Leaf herbivory varies between subtropical and tropical tree species due to different plant traits (phenology, morphology, nutrient status) (*Schuldt et al, 2012*) and insect consumers (*Schuldt et al., 2010*). However, the mean percentage of leaves attacked in this study was high (74% and 60% in deciduous and evergreen trees, respectively), especially for some subtropical trees such as *A. fortunei, C. axillaris, C. tibetana*, which exceeded 90%. This may be attributed to their higher density and frequency in this forest, which could affect the presence of herbivores and intensity of leaf damage. Considering species as random effects, there was no significant difference in percentage of leaf area removed between deciduous and evergreen species, which was inconsistent with Hypothesis 1. Some studies previously found that evergreen species tend to exhibit lower intensities of leaf herbivory than deciduous species in tropical forests (*Pringle et al., 2011*; *Silva, Espírito-Santo & Morais, 2015*), due to their higher toughness, lower water content, and higher C:N ratios (*Schuldt et al., 2010*).

In general, foliar secondary metabolites as specific defensive traits of plants could decrease leaf palatability (*Kessler & Baldwin, 2002*; *Kazakou et al., 2019*) and reduce leaf herbivore attack (*White & Whitham, 2000*). In our study, the percentage of leaf area removed was positively correlated with the concentrations of tannins, cellulose and N in leaves, and negatively correlated with the ratio of NSC:lignin, the concentration of hemicellulose and C, which was inconsistent with Hypothesis 2. Plants use multiple defense mechanisms to overcome insect pests, which can be constitutive or induced, and direct or indirect, depending upon the insect and the degree of insect attack (*Ahuja, Rohloff & Bones, 2011*; *War et al., 2012*). While the concentrations of tannins and lignin influence the palatability and physical traits of leves, estimating only tannins and lignin in leaves

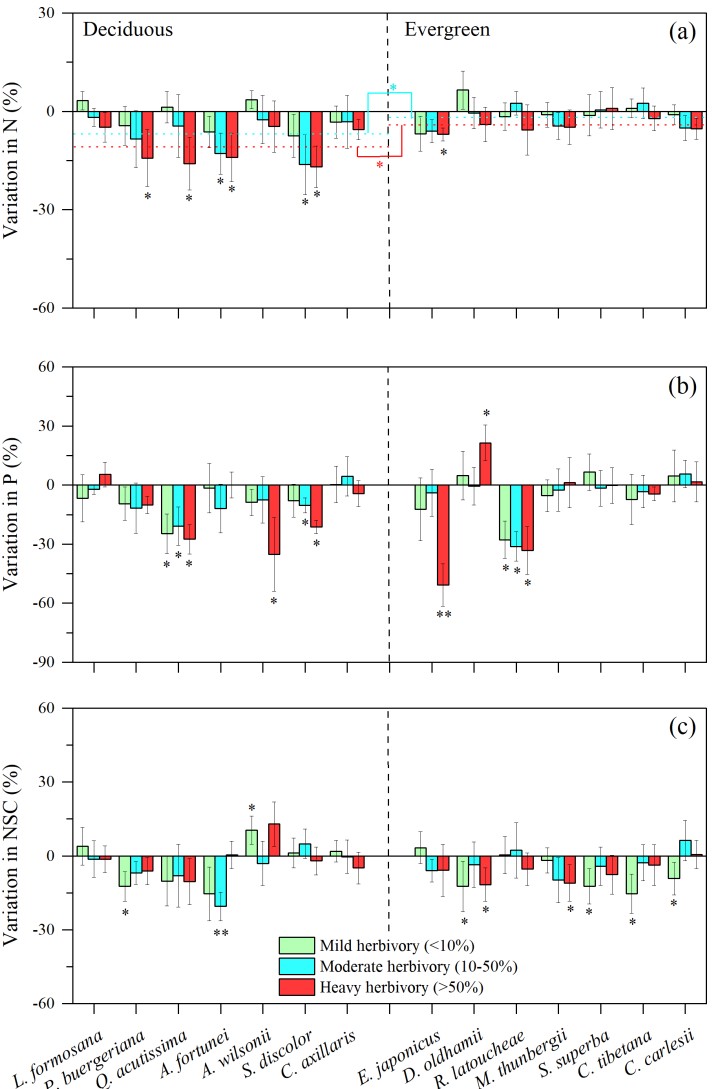

**Figure 2** **Variation in leaf N, P and NSC under three levels of insect herbivores for 14 subtropical tree species (mean ± SE, $n = 6$), presented as increased concentrations (positive values) or decreased concentrations (negative values) of leaf: (A) nitrogen (N), (B) phosphorus (P), and (C) Non-structural carbohydrates (NSC).** Herbivory intensities were classified as low (< 10%), medium (10–50%) and high (> 50%). Both deciduous and evergreen species were included: *Liquidambar formosana*, *Padus buergeriana*, *Quercus acutissima*, *Alniphyllum fortunei*, *Acer wilsonii*, *Sapium discolor*, *Choerospondias axillaris*, and *Elaeocarpus japonicus*, *Daphniphyllum oldhamii*, *Rhododendron latoucheae*, *Machilus thunbergii*, *Schima superba*, *Castanopsis tibetana*, *Castanopsis carlesii*. Dotted line indicates the mean leaf N responses to herbivores. Significant differences between deciduous and evergreen species, and significant deviations among species from zero are denoted as $^*P < 0.05$, $^{**}P < 0.01$.

may not represent the whole range of defense, and other defense mechanisms might play crucial roles in alleviating insect attack. Indeed, leaves with high concentrations of Si, high toughness, many trichomes, or waxy surfaces, might negatively affect herbivores by decreasing palatability and digestibility (*Yamawo et al., 2012*; *Alhousari & Greger, 2018*;

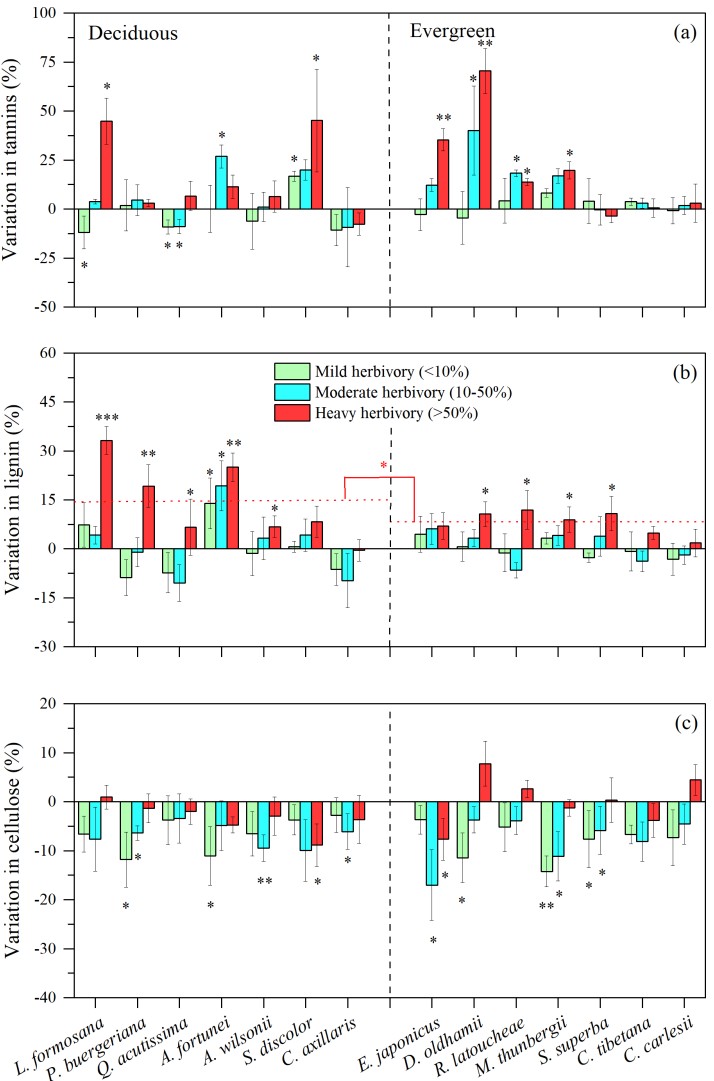

**Figure 3 Variation in leaf tannin, lignin, and cellulose under three levels of herbivory for 14 subtropical tree species (mean ± SE, $n = 6$), presented as increased concentrations (positive values) or decreased concentrations (negative values) of leaf: (A) tannin, (B) lignin, and (C) cellulose.** Herbivory intensities were classified as low (< 10%), medium (10–50%) and high (> 50%). Both deciduous and evergreen species were included: *Liquidambar formosana, Padus buergeriana, Quercus acutissima, Alniphyllum fortunei, Acer wilsonii, Sapium discolor, Choerospondias axillaris,* and *Phyllostachys pubescens, Elaeocarpus japonicus, Daphniphyllum oldhamii, Rhododendron latoucheae, Machilus thunbergii, Schima superba, Castanopsis tibetana, Castanopsis carlesii.* Dotted line indicates the mean leaf lignin responses to herbivores. Significant differences between deciduous and evergreen species, and significant deviations among species from zero are denoted as $^*P < 0.05$, $^{**}P < 0.01$.

*Chaudhary et al., 2018*). Moreover, there are various secondary compounds in plants, such as alkaloids, amines, flavonoids, *etc.* (*Wink, 2010*; *Kazakou et al., 2019*). Unfortunately, we did not measure the concentrations of other secondary metabolites, thus were not able to account for the influences of these other potential factors on the percentage of leaf area removed.

## Leaf response and defense to varying intensities of herbivory

We found that leaf herbivore attack was associated with the concentrations of nutrients and defense compounds, and there were decreased concentrations of nutrients and NSC, and increased concentrations of tannins and lignin for defense in some species, such as *P. buergeriana*, *A. fortunei*, and *S. discolor*, particularly under high herbivory, which supports Hypothesis 3. Here, the concentrations of defense compounds among plant species vary greatly, which may be a crucial factor in determining the intensity of herbivore attack (*Harborne, 1991*). Under conditions of low herbivore pressure, plant fitness would be maximized by investing little in defenses. In contrast, under conditions of high herbivore pressure, the fitness benefits of constitutive defenses could outweigh the fitness costs. However, other studies showed there were no significant responses of tannin production to leaf herbivory for species, possibly due to reliance on different defensive mechanisms, such as constitutive defenses including high toughness, many trichomes, or waxy surfaces (*Peeters, 2002*; *Onoda et al., 2011*). Plants will make trade-offs between growth and constitutive defense, because constitutive defense is relatively costly (*Stamp, 2003*).

In addition, there were increased concentration of lignin in most species, particularly under high levels of herbivory (>50% leaf area removed). Some studies found that more lignin was produced by leaves following insect herbivory, contributing to more leaf toughness, less palatability and digestibility (*Marler & Dongol, 2016*). It is also possible that lignin values were higher because the more palatable tissues of the leaf were largely consumed by insect herbivores, and higher lignin-containing veins remained (*Beck & Labandeira, 1998*). Moreover, the leaf defense response of some deciduous species (*L. formosana*, *P. buergeriana*, *A. fortunei*) was stronger than that of most evergreen species, particularly the increased concentrations of lignin. Long-lived evergreen leaves may invest more in construction and maintenance than leaves of deciduous species (*Chaturvedi, Raghubanshi & Singh, 2011*; *Eamus, 1999*; *Sobrado, 1991*). Notably, all defense responses may be different among the branches within a tree due to differential herbivory damage. Therefore, the measurements and the degree of leaf herbivory in this study may not reflect the actual level of herbivory and defensive response. For example, leaf chemistry values on average may be simply a product of some heavily damaged leaves with induced responses that elevate subsequent defenses, in combination with by low herbivory on the (better defended) remaining foliage.

N and P could be diverted or leached out after herbivory, thus, N and P could also be lower in damaged individuals, which could reduce herbivory due to decreasing concentrations of N and P. Previous studies have shown that NSC was reduced in herbivore-attacked plant tissues, because resources were diverted away from the site of damage and into storage tissues (*Newingham, Callaway & BassiriRad, 2007*; *Gómez et al., 2010*; *Quijano-Medina et al., 2019*). On the other hand, herbivore attack can greatly weaken plant photosynthesis due to leaf area loss and tissue damage (*Coley, 1988*; *Bilgin et al., 2010*; *Visakorpi et al., 2020*). It is also possible that plants will invest more in defense under leaf herbivore attack, instead of growth (*Herms & Mattson, 1992*). In our study, insect herbivory reduced cellulose for most species, which was in agreement with other studies (*Onoda et al., 2011*). Cellulose is a

major component of plant cell walls and can influence plant growth and stability (*Taylor, 2008*). It has been reported that the enzymes secreted by insect herbivores can destroy the physical tissues of plant leaves, especially in the plant cell wall, which is dominated by cellulose (*Schowalter, Hargrove & Crossley, 1986*; *Onoda et al., 2011*), making the leaves more susceptible to pathogen infection due to weaker physical defenses.

## CONCLUSION

Our study reveals a higher percentage of leaves damaged in deciduous than evergreen species in subtropical forests among a relatively small set of tree species, but no significant differences in the percentage of leaf area removed, as that may be affected by other factors such as plant density and insect community dynamics. While leaf herbivory measurements were positively correlated with the concentrations of tannin and negatively correlated with the ratio of NSC:lignin, we were not able to explain the percentage of leaf area removed, likely due to the production of other defense mechanisms. We should investigate the relationship between leaf herbivory and additional leaf traits (such as toughness, thickness, or other defense compounds) for more subtropical species in the future. In addition, we should explore the effects of plant density and insect community structure on insect herbivory and the percentage of leaf area removed. Collectively, our findings elucidate how leaves respond to varying intensities of herbivory, and we explore some of the underlying relationships between leaf traits and herbivore attack in subtropical forests.

## ACKNOWLEDGEMENTS

We appreciate Guiwu Zou for help with some data analysis.

### Funding

This work was supported by the National Natural Science Foundation of China (No. 42067050, 41807028, 32060319, 42267034); This research was supported by the Jiangxi Double Thousand Plan: jxsq2020101079; Graduate Innovation Fund of Jiangxi Province (YC2021-S361). Support for Carri LeRoy was provided by U.S. National Science Foundation grant DEB #1836387. The funders had no role in study design, data collection and analysis, decision to publish, or preparation of the manuscript.

### Grant Disclosures

The following grant information was disclosed by the authors:
National Natural Science Foundation of China: 42067050, 41807028, 32060319, 42267034.
Jiangxi Double Thousand Plan: jxsq2020101079.
Graduate Innovation Fund of Jiangxi Province: YC2021-S361.
US National Science Foundation: DEB #1836387.

### Competing Interests

The authors declare there are no competing interests.

## Author Contributions

- Xiaoyu Liu conceived and designed the experiments, analyzed the data, prepared figures and/or tables, authored or reviewed drafts of the article, and approved the final draft.
- Carri J. LeRoy analyzed the data, authored or reviewed drafts of the article, and approved the final draft.
- Guobing Wang performed the experiments, authored or reviewed drafts of the article, and approved the final draft.
- Yuan Guo conceived and designed the experiments, prepared figures and/or tables, and approved the final draft.
- Shuwang Song conceived and designed the experiments, prepared figures and/or tables, and approved the final draft.
- Zhipei Wang conceived and designed the experiments, authored or reviewed drafts of the article, and approved the final draft.
- Jingfang Wu performed the experiments, authored or reviewed drafts of the article, and approved the final draft.
- Fenggang Luan performed the experiments, authored or reviewed drafts of the article, and approved the final draft.
- Qingni Song analyzed the data, authored or reviewed drafts of the article, and approved the final draft.
- Xiong Fang analyzed the data, prepared figures and/or tables, and approved the final draft.
- Qingpei Yang analyzed the data, authored or reviewed drafts of the article, and approved the final draft.
- Dongmei Huang performed the experiments, prepared figures and/or tables, and approved the final draft.
- Jun Liu analyzed the data, authored or reviewed drafts of the article, and approved the final draft.

## Field Study Permissions

The following information was supplied relating to field study approvals (i.e., approving body and any reference numbers):

Field experiments were approved by the local government of Jiangxi Guanshan National Nature Reserve (project number: The National Natural Science Foundation of China, 32060319).

## Data Availability

The raw measurements are available in the tables, figures and Supplementary Files.

## Supplemental Information

Supplemental information for this article can be found online at http://dx.doi.org/10.7717/peerj.16350#supplemental-information.

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
