# Peer review of "Leaf defenses of subtropical deciduous and evergreen trees to varying intensities of herbivory"

_PeerJ, doi:10.7717/peerj.16350_

## Round 0.1 · original submission · Major Revisions

Thank you to the two expert reviewers for their thorough reviews.

Both reviewers found the manuscript to be generally well-written and that the authors got their points across. Both reviewers also mentioned that this is an interesting study and a generally good approach.

Both reviewers discuss the issue with rather small sample sizes in making the comparison between deciduous and evergreen trees. Reviewer 1 also provides a number of important points related to statistical approaches and analyses. And both reviewers discuss a need to rethink and restate some of the hypotheses.

In addition Reviewer 1 mentions a need to contextualize this work better by a better survey of the literature and use of more relevant citations.

I agree with all of these recommendations, and I feel that the level of revisions is major. As such I am recommending major revisions. Upon receipt of a revised manuscript and response/rebuttal, I expect that I will send this out for a second round of review – either to the same reviewers if they are amenable, or to new reviewers.

Sorry for the delay in processing this manuscript – finding reviewers was quite difficult. Thanks to the two reviewers for the fast turnaround once they received the manuscript.

Reviewer 1 ·

Basic reporting

The manuscript summarizes a rather large study and, overall, is quite well-written (with a few places where the wording could be improved) and structured with good clarity around three main hypotheses.

The references in the manuscript feel somewhat haphazard; some key citations are missing, while others chosen to support certain statements are relatively obscure and of secondary relevance. The field of plant defense theory is vast and can feel overwhelming, but there should be little difficulty in finding appropriate citations and strengthening the manuscript by invoking them.

Species names on x-axes in the figures should be italicized.

Experimental design

The manuscript by Liu et al. explores the differences in the levels of leaf herbivory, and the underlying foliar traits that may explain this variation (and respond to it), examining a small sample of subtropical tree species from southern China. This study follows a long tradition of research and hypotheses to predict and explain the differences in the types and levels of defense investment and associated ecological tradeoffs among trees in species-diverse forests. It focuses on contrasting deciduous and evergreen species, and fills an empirical gap in the relatively less studied region and biome. I particularly commend the authors for looking at a sizeable range of foliar traits and sampling a relatively phylogenetically diverse assemblage of trees. They report on a valuable dataset and there is great potential in learning from it.

My main concerns about the current version centre around three general areas that, in my opinion, require improvements or caveats in the form of a major revision, if feasible. The first criticism is the way the conceptual premise of the study is framed within a large body of literature on plant defense theory; the second criticism questions some of the rationale behind the hypotheses (see next section for my comments on both). Finally, the third issue focuses on acknowledging the limitations of the study design, and hence the interpretation of the findings and testability of the hypotheses:

Study design
i) Replication within species – while it’s difficult to increase the sample size for each species, given the analytical work, I worry 6 individuals may not be able to capture the spatial heterogeneity that is common for insect herbivory – which makes the comparison of deciduous vs. evergreen species more tenuous. It is also inadequate to examine predictive relationships between foliar traits and herbivory within a species – i.e. comparing variation among individuals of the same species – which may be a more appropriate level of asking some of the questions in this study, prior to contrasting deciduous and evergreen species.
ii) Without knowing more about patterns of herbivory, what can be concluded about the level of damage and the underlying factors from pooled leaves? For instance, moderate levels on average may be simply a product of some heavily damaged leaves, induced response that elevates subsequent defenses, followed by low herbivory on the (better defended) remaining foliage. It is unclear how this may have been taken into account.
iii) While it makes sense to divide the degree of herbivory into categorical bins, this could also reduce the ability to detect differences in other variables – for instance, mild (<10%) but non-zero herbivory may not be functionally very different from, say, 15-25% which already falls into the next (moderate) category.
iv) Data analysis deserves more elaboration. For instance, is individual included as a random factor (to avoid pseudoreplication)? Are leaf responses calculated by comparing pooled leaves of different levels of damage at the level of individuals?

Validity of the findings

1. Hypotheses and broader theory of plant defenses
The introduction should more explicitly include the concepts of constitutive and induced defense, plant defenses against specialist versus generalist herbivores, and the relevant key references. These distinctions have major implications for the theoretical predictions, as well as the inference about plant responses that can be made from observational studies without experimental manipulations (also see below). There is relatively little attention given to some of the fundamental concepts in plant defense theory, such as the growth differentiation balance hypothesis and related trade-off concepts (e.g. see Fine 2006), tolerance vs. resistance, etc. – although the study poses questions that are relevant to many of these topics. The authors do a better job mentioning some of these concepts at least very briefly in the discussion but it would help structure the premise of their study if outlined more clearly early in the paper.

2. Rationale
The variation in herbivory among individuals (let alone species) is likely to be explained by multiple factors, some of which are interacting or even change over time. In addition, without experimental manipulations it is not trivial to disentangle the causal relationships and feedbacks between foliar traits that herbivores use in their foraging versus those that herbivory alters in an induced/plastic fashion. For instance, the variation in foliar nitrogen may explain which branches receive greater damage… consequently, what may seem like a leaf response in nitrogen content – based on damaged leaves compared to those without damage – may have generated differing levels of damage in the first place. Therefore I struggle understanding how the authors infer what they discuss as causal relationships, specifically under Hypothesis 3. Ultimately, the study is very much correlational and exploratory in nature, and while the authors acknowledge these drawbacks in several places in the discussion, they still opt to ask questions and interpret their findings in ways that, in my opinion, are over-reaching the study design.

Additional comments

Other comments
1. More detail on the sampling would be helpful – for instance, were the individual trees and each of the four branches selected at random? In L137, please use clearer wording to indicate that for each species six trees were sampled.
2. Were all leaves on a branch (of what length? Standardized?) collected and included in the pooled analysis of the four branches, or was it a subsample? How were the leaves that were entirely consumed or prematurely abscised (due to damage) accounted for?

Reviewer 2 ·

Basic reporting

The manuscript by Liu et al. describes an analysis of different subtropical trees and their response to insect herbivory. The manuscript is in most parts well written and structured, and provides sufficient references to support their findings and interpretations.

Experimental design

The experimental design is very thorough with sample sizes sufficient to perform meaningful statistical analyses. The hypotheses are well designed, although I would suggest to eliminate hypothesis 3a and 3b and rather have a 4th one. I would also suggest to add some more information on the lignin, cellulose, and hemicellulose analysis, which would help the reader. All other methods are described in sufficient detail.

Validity of the findings

As must be expected, the results of the study do only allow for a limited interpretation of the results. While the sample sizes were sufficient for the statistical analyses, I urge caution in using it for a larger interpretation regarding the differences between deciduous and evergreen trees in general. Here, the relatively small sample size of only 7 species for each group is not enough for meaningful conclusions related to the comparison that was the major focus of this study. This is also reflected in the results, which, while providing some cues, do not provide any evidence for significant differences between the two groups. This is a general problem with studies like this. Adding one more species to either group can alter the overall result. Therefore, I would recommend to avoid any general conclusions.

Additional comments

There are a few minor issues that need to be resolved.
Line 62/63, all plant groups host numerous insect herbivores, not just the ones presented in this manuscript.
Line 69, should be 'is' and not 'was".
Line 80, should be 'compared'.
Line 93, should be 'phenols'.
Line 123, should add '...plot, and were permitted...'.
Line 152, 30 min at 105ºC does not just surface sterilize, but rather kill everything.
Line 155, should be 'leaf herbivory'.
Line 166, NSC were already introduced before.
Line 182, how was leaf powder extracted? Needs more detail.
Line 289, tannins are not just a physical defense, but also a chemical.
Line 295, amino acids are not secondary metabolites.

---

## Round 0.2 · Minor Revisions

Thanks to the co-authors for their revised manuscript and their response to the previous reviews. Both previous reviewers have re-reviewed the revised version, and I am thankful for their expertise that has helped to improve the paper. Both reviewers feel that the paper is ready or almost ready to be published.

Reviewer 1 provides a few more suggestions for minor revisions that will strengthen the paper more prior to publication. As that reviewer notes, the suggested changes are to overall wording, but should make the message stronger and clearer. Following this final revision, I anticipate that the manuscript will be ready for publication.

I would like to again thank the reviewers for their excellent reviews throughout this process.

Reviewer 1 ·

Basic reporting

The authors have done a good job improving the introduction and providing a clearer conceptual background to their hypotheses. Likewise, the discussion is more balanced and offers a less over-reaching interpretation of their results (but see below). My suggestions for the previous version were satisfactorily integrated in the revisions. Most of my comments on the revisions focus on the wording, but I do consider them to be significant in helping to strike the right tone for how the study is framed.

Experimental design

With respect to the comparison of foliar traits under varying levels of herbivory – I wonder if an additional, and perhaps more appropriate, comparison should be between leaves with low (but non-zero) and high levels of damage. A leaf that shows no signs of herbivory may have simply been missed, but a leaf with some minor damage could imply that the insect found and tasted it, but was deterred/discouraged by the levels of defense or nutrient quality, and went on to feed on another, better leaf. Therefore it would be an interesting, more focused contrast for the authors to consider adding to strengthen their argument, even if it still cannot fully separate pre-existing versus induced differences in defensive traits.

Validity of the findings

It would be helpful to mention in the last paragraph of the introduction that the study is correlational (in linking damage to putative responses) and that it is using ambient (natural) levels of herbivory, rather than manipulating herbivory.

My main criticism is that the wording in the Results and Discussion still occasionally implies a more causal relationship between herbivory and leaf traits. For instance, wording such as “herbivore attack decreased the concentration of…” (L354-355) implies causality where it may not be warranted without an appropriate study design. I suggest using wording such as “was associated with”.

I would also encourage the authors to consider an alternative term than “leaf responses” as their study design cannot truly disentangle foliar traits/levels prior to versus following herbivory, since only varying intensities (i.e. defense outcomes) of herbivory are measured, without knowing the pre-existing differences.

Additional comments

L67-68 – odd wording in this sentence; maybe delete as the next sentence gives a similar but clearer message
L71-72 – this statement should have a reference or two, and it would be a good first sentence of a new paragraph (as the focus shifts on traits)
L155 – how were these species selected? – information on this would help clarify how representative the findings and conclusions are for this forest type.

Reviewer 2 ·

Basic reporting

Since the last review the manuscript has improved significantly. The authors have addressed all of my concerns. I find the manuscript informative, and have nor further suggestions for improvements.

Experimental design

The experimental design is clearly described. I have no further issues with those.

Validity of the findings

The authors have addressed my main concern by removing the generalization of their data. It must be clear that a study, which only includes 14 species, cannot be used for generalization, but do still provide interesting data on differences between the groups within their limited spacial scope. I have no further concerns.

Additional comments

I have no further comments.

---

## Round 0.3 · accepted · Accept

Through several rounds of review, the authors have now adequately addressed all of the concerns brought up by the reviewers. The review process has been constructive in improving this manuscript with each iteration reading better than the previous one. The authors have done a good job of tracking and explaining changes. I am satisfied that it is ready to be published in PeerJ in its current form and will make a useful contribution to the literature. Thank you to the reviewers and the co-authors for a good review process.